# Perturbation by Antimicrobial Bacteria of the Epidermal Bacterial Flora of Rainbow Trout in Flow-Through Aquaculture

**DOI:** 10.3390/biology11081249

**Published:** 2022-08-22

**Authors:** Hajime Nakatani, Naoki Yamada, Naoki Hashimoto, Fumiyoshi Okazaki, Tomoko Arakawa, Yutaka Tamaru, Katsutoshi Hori

**Affiliations:** 1Department of Biomolecular Engineering, Graduate School of Engineering, Nagoya University, Furo-cho, Chikusa, Nagoya 464-8603, Aichi, Japan; 2Department of Life Sciences, Graduate School of Bioresources, Mie University, 1577 Kurimamachiya, Tsu 514-8507, Mie, Japan

**Keywords:** antimicrobial substances, bacterial flora, fish skin, flow-through aquaculture, microbial ecology, rainbow trout

## Abstract

**Simple Summary:**

The epidermis and epidermal mucus layer of fish function as a first barrier against waterborne pathogens, while also providing niches for symbiotic microorganisms that can be beneficial to the host’s health. Controlling the composition of the fish skin microflora is important in pathogen management in aquaculture; however, our understanding of the characteristics of this microflora is limited. To elucidate the characteristics of fish skin bacterial flora, we examined the epidermal mucous of rainbow trout reared in flow-through aquaculture as an experimental model with environmental perturbations. Our experimental data indicate that some specific bacteria with antimicrobial activity enter from the natural environment and affect compositional changes in the skin mucous bacterial community by disturbing and occupying it. This perturbation by antimicrobial bacteria can “remodel” the fish skin bacterial flora and thereby affect the host′s health. This study provides novel information on factors influencing the composition of fish skin bacterial flora, which is applicable for controlling fish disease by using beneficial bacteria in aquaculture.

**Abstract:**

The bacterial flora of the epidermal mucus of fish is closely associated with the host’s health and susceptibility to pathogenic infections. In this study, we analyzed the epidermal mucus bacteria of rainbow trout (*Oncorhynchus mykiss*) reared in flow-through aquaculture under environmental perturbations. Over ~2 years, the bacteria present in the skin mucus and water were analyzed based on the 16S rDNA sequences. The composition of the mucus bacterial community showed significant monthly fluctuations, with frequent changes in the dominant bacterial species. Analysis of the beta- and alpha-diversity of the mucus bacterial flora showed the fluctuations of the composition of the flora were caused by the genera *Pseudomonas*, *Yersinia*, and *Flavobacterium*, and some species of *Pseudomonas* and *Yersinia* in the mucus were identified as antimicrobial bacteria. Examination of the antimicrobial bacteria in the lab aquarium showed that the natural presence of antimicrobial bacteria in the mucus and water, or the purposeful addition of them to the rearing water, caused a transition in the mucus bacteria community composition. These results demonstrate that specific antimicrobial bacteria in the water or in epidermal mucus comprise one of the causes of changes in fish epidermal mucus microflora.

## 1. Introduction

The skin of fish is exposed to many active microorganisms present in the water; thus, their epidermis and epidermal mucus layer function as a first barrier to physically and immunologically protect the fish from waterborne pathogens [1,2,3]. In addition to the host defense systems, symbiotic microorganisms in the fish epidermal mucus layer can benefit the host’s health by protecting against pathogenic infections [3,4,5,6,7,8,9]. Accordingly, the importance of the composition of the fish epidermal mucus microflora for disease control in cultured fish has often been mentioned, yet studies characterizing the fish skin microflora are limited [10]. Several basic studies on this topic have demonstrated that the composition of fish skin microflora is distinct from the bacterial flora in the water [11,12]. However, the composition of the fish skin microflora is susceptible to changes in environmental conditions or the composition of the microbes present in the water [11,13,14]. Fluctuations in the bacterial community composition have been linked to changes in pH [15], salinity [16,17], dissolved oxygen and nutrients [18], as well as to fish handling [19], the diversity of the microbes, and the presence of infectious pathogens in the water [20,21]. Fluctuations in the fish skin microflora’s composition from environmental perturbation would affect the host’s susceptibility to infection disease if the composition of the flora influences the host’s defense system [4,21].

To understand the relationship between the compositional differences in fish skin microflora and the susceptibility to percutaneous infection, we established a percutaneous infection model with zebrafish (*Danio rerio*) and a fish pathogen *Yersinia ruckeri*, which causes enteric redmouth disease in salmon and trout [22]. Using this model, we demonstrated fluctuations in the composition of the epidermal bacterial flora following temperature shifts in the culture water and the stress of skin injury, as well as the disruption of the epidermal microflora by antibiotics, which promoted the colonization and establishment of *Y. ruckeri* on the fish skin [21]. These results indicate that fluctuations in fish epidermal bacterial flora can promote the invasion and establishment on the epidermis of pathogens and other microorganisms in the water.

To explore the main factor causing changes in the composition of the fish epidermal mucus bacterial flora in aquaculture, this study examined changes in the composition of the epidermal mucus bacterial flora of rainbow trout, farmed in a flow-through system with river water, with perturbations from the outer environment, over a period of two years. The skin microflora of fish reared in a lab environment was also examined as a condition of an environment without external perturbations. Furthermore, we investigated involvement by specific bacterial groups that trigger bacterial interactions, as a factor capable of causing large changes in the epidermal mucus flora under aquaculture conditions.

## 2. Materials and Methods

### 2.1. Manipulation of Bacterial Strains, Cultivation of Fish Skin Bacteria, and Screening

*Yersinia ruckeri* NVH 3758 was isolated from rainbow trout with enteric redmouth disease in Norway and provided to us by Dr Dirk Linke, University of Oslo, Norway; *Y. ruckeri* NCTC 12266 was purchased from the National Collection of Type Cultures (NCTC) in the United Kingdom. For the artificial infection experiments or for the maintenance of them, these *Yersinia* strains were cultured in LB medium (Miller) at 28 °C for 24 h, with shaking at 115 rpm. *Aeromonas hydrophila* NRIA14, *Vibrio anguillarum* NRIA83, and *Vibrio ordalii* NRIA90 were provided by the Japan Fisheries Research and Education Agency. *A. hydrophila* was maintained in nutrient broth, and the *Vibrio* species were maintained in Marine broth 2216 (Becton, Dickinson and Company, Franklin Lakes, NJ, USA).

To isolate bacterial strains from the intrinsic skin microflora of rainbow trout, each sample of suspended skin mucus in sterile water was spread on a nutrient agar medium, enriched Cytophaga agar medium, modified Zobell 2216E agar medium (0.8% NaCl), and LB agar medium or tryptone soya broth agar medium, then incubated at either 20 °C or 15 °C for 2 to 4 days to grow bacterial colonies.

To determine the growth-inhibition activities of fish epidermal mucus bacteria, the fish pathogens *A. hydrophila* NRIA14, *V. anguillarum* NRIA83, *V. ordalii* NRIA90, *Y. ruckeri* NVH 3758, and strains isolated from the fish skin (see above) were pre-cultured in a liquid medium overnight, and the bacteria were then used for cross-streak plating. The cross-streak test was performed as described previously [23]. The streak cultures were incubated at 20 °C or 15 °C for two days. The 16S rDNA of bacteria that showed an antagonistic effect against the pathogen’s growth was amplified, and the V1–V3 region was sequenced; the closest bacterial species was identified by a BLAST search of 16S rDNA sequences. The primers used for 16S rDNA amplification and sequencing of the V1–V3 region are listed in Appendix A.

### 2.2. Next-Generation Sequencing of 16S rDNA Amplicon Libraries

The skin bacterial flora of rainbow trout were collected using an Alpha TX swab (TX761, Texwipe, Kernersville, North Carolina, USA), and the swabs with the mucus sample were stored in 80% ethanol in the field, then stored at 4 °C until DNA extraction (Appendix A). Water samples (~500 mL) were collected from each culture pond at the water intake (Appendix A) and kept at 4 °C until filtration. Filtration with suction through a mixed cellulose membrane filter (0.45 μm pore size, 47 mm diameter; ADVANTEC, Tokyo, Japan) was performed to collect the microorganisms in the water, and the microorganisms trapped on the membrane filter were stored at −30 °C until DNA extraction. The genomic DNA extraction from the sample was performed as described previously [21]. The NucleoSpin Tissue kit (Takara, Otsu, Japan) was used to extract and purify genomic DNA from the samples, according to the protocol for recovering bacterial genomic DNA from difficult-to-lyse bacteria such as Gram-positive bacteria. Bacterial 16S rDNA amplicon libraries were prepared from the extracted genomic DNA for DNA sequencing using iSeq 100 (Illumina, San Diego, CA, USA), according to the Illumina manual (16S Metagenomic Sequencing Library Preparation, Part #15044223 Rev. A), as previously described, but with slight modifications. The 16S rDNA V1–V2 region was amplified from the extracted genomic DNA by 30 cycles of PCR. The annealing temperature was set at 51 °C. The primer sequences for specific amplification of the V1–V2 region, with overhang sequences for adaptor extension, are shown in Appendix A. The prepared library concentration was finally adjusted to 50 pM for analysis by the iSeq100 Sequencing System (guide 1000000036024 v03 JPN; Illumina). PhiX Control v3 (Illumina) was added to the library to enhance sequence data quality.

### 2.3. In Silico Analysis Based on 16S rDNA Amplicon Sequencing

The CLC Genomics Workbench (Qiagen, Tokyo, Japan) with the Microbial Genomics Module was used for bacterial flora analysis of 16S rDNA sequences obtained by iSeq100. The analyses were performed using sense reads (approximately 150 bp) of the pair-end fastq, because the merged sequences were not adequately obtained. Sequence data were imported into the software, and the reads were trimmed, based on the length (140–160 bp) and quality score and the number of ambiguous nucleotides, using the “Trimming Sequence” tool. To classify reads according to the experimental conditions, metadata files were created that included parameters for sample name, sampling month, sampling year, age, water temperature deviation, and the information on the aquaculture ponds.

Reference-based operational taxonomic unit (OTU) clustering was performed using the “OTU Clustering” tool, as described previously [21]. The SILVA 16S rDNA database v132 (https://www.arb-silva.de (accessed on 16 October 2021)) was used as the reference database with a similarity threshold of 97%. The abundance tables including OTU data were exported as a Microsoft Excel spreadsheet (Appendix A). To identify specific bacterial species from the read sequences, we searched for sequences similar to the read sequences in the NCBI 16s rDNA database using BLAST.

Analyses of α-diversity were performed using OTU abundance tables [21]. Heatmaps were generated using the “Create Heat Map for Abundance Table” tool [21]. An abundance table was generated from the sequence data of the samples to be compared, and the heat map was created using the abundance of the top-100 OTUs with at least 100 reads.

Alpha-diversity indices were calculated using specific tools in the CLC Genomics Workbench [24,25]. To generate rarefaction curves for the Shannon index of α-diversity, the range of the subsampling depth was set from 1 to 20,000 reads, the number of different depths for sampling was set to 100, and the number of replicates for each depth was set to 100. The results of rarefaction curves were exported as Microsoft Excel spreadsheets (Appendix A), and the results at the subsampling depth (2400 reads) are presented as box plots.

Mapping of the reads obtained from each sample to the 16S rDNA sequences of bacteria that produce antibacterial substances was performed as follows. The trimmed reads of 16S rDNA from the samples were further filtered with the “taxonomic analysis” tool. The reads that matched the SILVA v132 reference files were extracted, and sequences with <90% similarity to any of the bacterial 16S rDNA sequences were discarded. The rarefaction analysis for the Chao1-bc index of alpha-diversity was performed with the range of the subsampling depth set from 1 to 20,000 reads, the number of different depths for sampling set to 100, and the number of replicates for each depth set to 100 (Appendix A). Based on the result, the random sampling of the reads (3000 reads from each skin mucus sample; 5000 reads from the laboratory rearing water samples; and 20,000 reads from each water sample from the fish farm) was performed from the filtered reads. The sampled reads from the same collection time were then merged. The merged sequences were mapped to the sequences list of 16S rDNA containing the V1–V2 region obtained from the antimicrobial bacteria, and finally, the mapped reads to the sequence of each bacterium were counted (Appendix A). The proportion of the mapped reads is expressed as a heatmap.

### 2.4. Maintenance and Handling of Rainbow Trout in Fish Farm and Laboratory Aquarium

The study of the fish epidermal mucus bacterial flora was conducted at the Uren fish farm (lat. 35°08′53″ N, long.137°33′12″ E), where the flow-through aquaculture system is. An overview of the rearing ponds at the Uren farm is shown in Appendix A. The rearing water used in the ponds is obtained from a natural river and is supplied to three rearing ponds A, B, and C. Rearing Pond A is directly filled with river water, while Ponds B and C receive river water and the rearing water from Ponds A and B, respectively. The new fish (about 2 months old) were reared in Pond A at least for one month, and transferred to Pond B or Pond C when the new ones entered into Pond A. For sampling of the fish skin mucus, three samples of microorganisms in the epidermal mucus of fish were collected from fish rearing in Ponds A, B, and C per one sampling, respectively. The fish were captured from the ponds by netting and then held by hand while wearing latex gloves as a sample of epidermal mucus was removed by gently scraping with a swab along the side of the fish. The river water and rearing water from Ponds A, B, and C (500 mL) were also collected for samples of microorganisms in the water.

For the experiment conducted in the laboratory aquarium, Rainbow trout (average weight 12.5 g, approximately the same size) were transferred from the fish farm to a laboratory aquarium (100-L, custom-made) fit with a water recirculation and filtration system (Eheim Ecco Comfort 2232; EHEIM GmbH & Co. KG, Backnang, Germany) and water chiller (ZC-500α; Zensui, Osaka, Japan) and maintained at a rearing density of 5.2 g L^−1^. The water temperature was kept at 15 °C, and the water was changed every 2 days; regular feeding was performed during the acclimation period of ~2 weeks. During the experiments, water changes and feeding were suspended for ~1–2 weeks. The mortality rate was almost 0% during acclimation to the rearing environment and the experimental period, with the exception of fish that jumped out of the aquarium. The sampling of the fish and rearing water was performed from each aquarium, and the water quality (i.e., measures of pH, NO_3_-N, DO, water temperature) was checked several times during the experiment (Appendix A). The samples were temporarily stored at −30 °C (for fish samples) or 4 °C (for water samples), until they were further processed for analyses of bacterial flora.

The addition of the antimicrobial bacteria *Pseudomonas* and *Yersinia* to the aquarium was performed as follows. Bacterial suspensions of *Y. ruckeri* NCTC 12266 or *P. marginalis/rhodesiae* KH-RT 1, *P. koreensis* KH-RT 2, and *P. protegens* KH-RT3 and KH-RT5 at OD_600_ (optical density measured at a wavelength of 600 nm) of 1.0 in rearing water were added to the aquarium so that the final OD_600_ of each species in the rearing water was reduced to 0.0025. The first exposure to the bacteria for 4 days was followed by a second exposure to the *Pseudomonas* species.

## 3. Results

### 3.1. Changes of the Epidermal Mucus Microflora of Farmed Rainbow Trout in a Flow-Through System

To explore the characteristics of the epidermal mucus microflora of a farmed salmonid in a flow-through aquaculture system using natural river water (Appendix A), the composition of the epidermal mucus bacterial flora on the epidermis of rainbow trout was analyzed over the course of nearly 2 years, from April 2019 to November 2020. Sampling of the skin mucus was performed at intervals of one or two months, and the bacterial composition of the skin mucus was analyzed by comprehensive sequencing of 16S rDNA amplified from the samples. A total of 3,266,490 of 5,111,721 reads were classified into 20,165 OTUs by reference-based OTU clustering. The bacterial compositions of skin mucus shown by the stacked bar chart (Figure 1) demonstrated that the composition of the bacterial flora changed in intervals of several months or at the time when new fish were added to the pond. The composition of the mucus layer microflora of individual fish at the same sampling times (sampled in the same month) was relatively similar regardless of the rearing batch, except for the samples from the months of June 2019 and March and May 2020. In some months, the proportions of specific bacterial genera, namely *Yersinia*, *Vibrio*, *Flavobacterium*, *Pseudomonas*, *Moraxella*, *Acinetobacter*, *Methylobacterium*, and/or *Aeromonas*, were relatively high in the epidermal mucus.

Some of these genera include known fish pathogens [22,26,27,28,29,30]. Actual bacterial coldwater disease caused by *Flavobacterium* [29] was observed in March and May in 2020 (personal communication with farmers), and the outbreak corresponded to the increased abundance of the OUT closely related to *Flavobacterium psychrophilum* in the epidermal mucus of some fish (Appendix A). The data demonstrated that the fish epidermal bacterial flora often fluctuated in the flow-through system, and the occupation of the epidermal flora by specific bacteria, including fish pathogens, sometimes occurred.

### 3.2. Changes in Abundances of Epidermal Mucus Bacteria Independent of the Abundance of Bacteria in Water and Other Conditions of Sampling

To examine the cause of fluctuations in the epidermal microflora of fish in the aquaculture system, we next analyzed the relationship with the bacterial flora in the rearing water and river water, which could potentially influence the composition of the mucus layer bacteria. The compositions of the microflora of the epidermis, rearing water, and river water samples, by month, were clustered based on the abundance of the OTUs that frequently appeared in the samples (top-100 OTUs). The results showed that the bacterial flora composition of the rearing water, river water, and epidermal mucus sampled in the same month usually fell into different clusters and the similarity among them was low, even though the compositions of some epidermal mucus flora and that of water microflora were similar in some months (Figure 2). Furthermore, the proportions of different bacterial genera in the mucus were independent of those in the water, whereby specific bacteria occupied the mucus and fluctuations occurred specifically in the mucus flora.

We further examined the similarity of the bacterial species between the skin mucus and water samples through beta-diversity analysis of the samples calculated by the unweighted UniFrac distance. The data were visualized by three-dimensional principal coordinates analysis (3D-PCoA). The results showed that the mucus bacteria formed a distinct group from the bacteria in water. However, the groups of bacteria in the fish skin mucus and in the water were not clearly separated, but formed a continuous group (Appendix A). The abundance in river water of the bacteria OTUs that most occupied the mucus was further analyzed. We found that the proportions of those OTUs in river water were ordinarily very low, though the proportions increased in some months. The proportion of OTUs, except for *Yersinia* and *Acinetobacter*, did not always show an increase at the time of their occupation of the fish skin mucus (Appendix A; Figure 1). These results suggest that the mucus flora OTUs or specific OTUs occupying the fish skin originated from the river water; however, the abundance of a bacterial genus in the water did not always affect its abundance in the mucus.

The clusters of OTUs obtained from epidermal mucus formed groups representing mainly samples from the same month and year (Figure 2). The categorization of the epidermal mucus microflora was independent of the different sampling ponds (Pond A, B, or C), water temperature range (2–18 °C), and age of the fish (2–14 months) (Appendix A), suggesting that these factors did not directly determine the composition of the fish skin bacterial flora in this aquaculture facility.

### 3.3. Specific Bacterial Species Involved in Fluctuations of the Composition of Skin Bacterial Flora in Farmed Rainbow Trout

To explore a major factor that affects the epidermal mucus flora of rainbow trout in flow-through aquaculture, we compared the similarity of the bacterial flora of individuals by β-diversity analysis using the weighted UniFrac distance [31]. This analysis was performed among all individual rainbow trout samples collected in 2019 and 2020, and the data are visualized by 3D-PCoA (Figure 3A).

The fish with a different composition of the epidermal mucus bacterial flora from that of the main group (Figure 3A, blue dotted circle) were mainly samples collected in 2020 (Figure 3A, red dotted circle). The epidermal mucus bacterial flora of these fish was dominated by *Vibrio* spp., as seen in Figure 1. The occupation by *Vibrio* spp. in 2020 continued from January to August (Figure 1), and the flora from these fish formed a subgroup (Figure 3A, red dotted circle). Further examination by β-diversity analysis was performed with the samples from the years 2019 and 2020 separately. The results showed that fish with more clearly changing bacterial flora (Figure 3B, red dotted circle) than the changes seen seasonally (Figure 3B, yellow dotted circle) were extracted in each year. For the year 2019, the samples depicting occupation by *Yersinia* spp. (September and December) and *Pseudomonas* spp. (June) were isolated as different groups from the main group (Figure 3B, red dotted circle). For 2020, the samples mainly occupied by *Vibrio* spp. belonged to the main group, and the samples occupied by other bacteria were isolated as specific groups: flora samples dominated by *Flavobacterium* spp. and *Pseudomonas* spp. in March and May and *Acinetobacter* spp. in June and the samples without *Vibrio* spp. in September and November were separate from the main group (Figure 3B red dotted circle). A small change of the flora was also observed in April 2020 and August 2019 because of the addition or exchange of fish in the pond. These results indicated that specific bacterial species, which temporarily occupied the epidermal mucus, were greatly involved in the fluctuation of the composition of the fish epidermal bacterial flora in the flow-through aquaculture system.

To find out which bacteria especially influenced the fluctuation of the mucus microflora of the rainbow trout, we next analyzed changes in Shannon’s entropy as an α-diversity index of the evenness of the bacterial species in the epidermal mucus of each sample (Figure 4). “Occupation” and “disturbance” of the epidermal bacterial flora will affect the uniformity of the proportion of each bacterium present. We defined “occupation” as a state in which the proportions of the specific bacteria in the epidermal bacterial flora are increased greatly in many fish and “disturbance” as a state in which the abundance of the bacterial population in the flora differs significantly among individual fish, depending on the degree of occupancy and the influence of the occupied bacteria in this study. A low value and variance of the index would indicate that occupation of the specific bacteria, and a high variance of the value among fish would indicate disturbance of the flora.

During the nearly two years of study, disturbance periods were observed in June 2019 and March 2020 (Figure 4, green triangles); periods of significant occupation were observed in September and December 2019 (Figure 4, red triangles). The notable bacterial genera in these months were *Pseudomonas*, *Yersinia*, and *Flavobacterium* spp., which suggests that these genera, in particular, are involved in the occupation or disturbance of the epidermal mucus of rainbow trout in this fish farm.

### 3.4. Involvement of Specific Antimicrobial Bacteria in Perturbation of the Fish Epidermal Mucus Microflora

Our previous study showed that occupation of the epidermal mucus bacterial flora by specific bacteria, such as waterborne pathogens, can be triggered by acute temperature shifts, disruption of the flora by antibiotics, and the stress of injury [21]. Therefore, we hypothesized that bacteria that produce antimicrobial substances are a main cause of the occupancy and fluctuation of the fish epidermal mucus bacterial flora; hence, we attempted to isolate and identify antimicrobial bacteria present in the epidermal mucus of rainbow trout, using the cross-streak method using the samples collected during one year, from April 2019 to March 2020. We isolated any colonies showing growth inhibitory activity against any of the fish epidermis-related pathogens, namely *Yersinia ruckeri*, *Vibrio ordalli*, *V. anguillarum*, and *Aeromonas hydrophila*, as model fish skin bacteria and obtained 48 bacterial colonies from the fish in flow-through aquaculture, which was analyzed by bacterial flora, and 2 colonies from the fish in another fish farm (Appendix A). Of these 48 clones from the flow-through system, 37 clones (77.1%) were *Pseudomonas* spp.; the remaining colonies contained 4 clones of *Y. ruckeri/massiliensis* (8.3%), 3 clones of *Lactococcus* spp. (6.3%), and 1 clone each of *Acinetobacter* sp. and *Chryseobacterium soldanellicola*, and two clones were unidentified (Table 1).

Based on sequences of the 16S rDNA V1–V2 region of the isolated bacteria, we investigated how many of the sequenced reads from the samples matched the 16S rDNA sequences of antimicrobial bacteria. The analysis was performed with reads randomly sampled from each skin mucus sample and rearing water sample. As a result, reads assigned to *Pseudomonas* spp., *Yersinia* spp., and *Chryseobacterium soldanellicola* were detected. Relatively high proportions in the epidermal mucus samples from June, September, and December 2019 and March 2020 were matched to 16S rDNA sequences of bacteria that produce antimicrobial substances: *P. parafulva*, *P. koreensis*, and *Y. ruckeri* (Figure 5). This showed that the proportion of antimicrobial bacteria on the fish skin was high when perturbations to the epidermal mucus flora were occurring. Correlation between the abundance of the antimicrobial bacteria on the epidermal mucus and that in the rearing water was also analyzed, but the correlation did not seem to be clear (Appendix A).

The proportion of these antimicrobial bacteria in the rearing water was also analyzed. The results showed that the antimicrobial bacteria assigned as *Pseudomonas* spp. also existed in the rearing water, and their abundance in the water often increased. Increases in the proportion of antimicrobial bacteria in the water did not always coincide with the timing of increased abundance in the epidermal mucus; however, their proportion in the water tended to increase before the occupation or disturbance of the fish skin mucus.

### 3.5. Transition of the Epidermal Mucus Bacterial Flora in a Stable Environment Was Linked to Antimicrobial Bacteria

To evaluate the change of the composition of the epidermal mucus bacterial flora under stable conditions (meaning separate from external environmental influences), we reared rainbow trout in a laboratory aquarium with a recirculating system, then observed any transition of the epidermal mucus bacterial flora. The rainbow trout for these experiments were transferred from the same fish farm where the field sampling was performed to the aquarium at the time when each trial was performed and acclimated there for at least 2 weeks. The bacterial flora of the skin mucus and in the rearing water and the proportions assigned as antimicrobial bacteria were then analyzed.

The initial epidermal bacterial composition of the fish was different between the trials because the fish were transferred from the fish farm in different months (Figure 6A). In two of the three trials, a transition of the epidermal mucus flora was observed during the observation period (1 week), whereas no transition was observed in the other trial (Figure 6A,B). In the rearing water, compositional changes of the bacterial flora were observed mainly in Exp. 1 from Day 0 to Day 8, and the change of the proportion of *Pseudomonas* was observed in other experiments (Figure 6A). In Experiment 1 (Exp. 1), a moderate transition of the flora was observed. The antimicrobial bacterium *Pseudomonas* was detected mainly in the water on the first day of testing, and the proportion of *P. protegens* was high as of the last day. The proportion of antimicrobial bacteria in the skin mucus was not high on the first day, but *P. parafulva* and *Y. ruckeri* increased from the middle to last days of the period. In Experiment 2 (Exp. 2), a large transition of the flora was observed: the proportions of *P. parafulva* and *P. koreensis* in the skin mucus layer were high on the first day, and the proportion of *P. protegens* in the rearing water increased midway through the experimental period. In Experiment 3 (Exp. 3), a transition of the flora was hardly observed: the proportion of antimicrobial bacteria was low in both the mucus and the water on the first day, and thereafter, the proportion remained low in the mucus, although their proportion in the rearing water temporarily increased as of the middle day of the experiment (Figure 6C).

The parameters relating to water quality (water temperature, dissolved oxygen, pH, nitrate nitrogen) changed slightly over the period of each experiment, but these changes were considered to be barely related to the transition of the bacterial flora composition in such a short period (Appendix A).

The results of Exp. 1–3 showed that a generally high proportion of antimicrobial bacteria was detected in the mucus and in the rearing water once the transition of the flora was observed during the experiment. Besides, the results suggest that their increase in the water did not always immediately influence the bacterial composition in the skin mucus, though the increase of the proportion of those bacteria in mucus always influenced the composition of the bacteria in the flora.

### 3.6. Transition of the Epidermal Mucus Bacterial Flora of Rainbow Trout Owing to Antimicrobial Bacteria in the Water

We next examined the potential effect of antimicrobial bacteria in the rearing water on the transition of the epidermal mucus bacterial flora and the degree of subsequent increases in the antimicrobial bacteria in fish skin mucus. To test which bacteria could be added to the rearing water, we chose strains of three species of *Pseudomonas* isolated from skin mucus and belonging to three clades on a phylogenetic tree (Appendix A). These were *Pseudomonas marginalis/rhodesiae* KH-RT1, which is rare in rainbow trout aquaculture, but is sometimes found in water; *P. koreensis* KH-RT2, often found both in the fish skin mucus and in water; and *P. protegens* KH-RT3 and KH-RT5, found mainly in water (Figure 5 and Figure 6C). These *Pseudomonas* strains exhibited wider antimicrobial activity against fish-related bacteria than *P. parafulva* N10L2-15, which is found mainly in fish skin mucus (Figure 5 and Figure 6C; Appendix A). We also chose a trout pathogen, *Y. ruckeri* NCTC12266, found in water that could adhere to the epidermis and increase its proportion in the epidermal bacterial flora when it is disturbed (Figure 6A) [21]. These bacteria were added to the water, and then, samples were collected for the observation of the transition of the epidermal mucus flora, as indicated in Figure 7A. Five bacterial strains, including four *Pseudomans* strains (KH-RT1, KH-RT2, KH-RT3, and KH-RT5) and one *Yersinia* strain, were added at the first dose. Four pseudomonas strains were added again at the second dose. *Pseudomonas* strains were added to the water twice to mimic the regular increase of them in the rearing water of the fish farm (Figure 5 and Figure 7A).

Three days after the first addition of *Y. ruckeri* and *Pseudomonas* to the rearing water, the bacterial flora changed from the initial composition, but the increase in the proportion of them in the epidermal mucus was slight. No significant change of the composition of the flora was observed until 4 days after the second addition of *Pseudomonas*, though the proportions of *Yersinia* and *Pseudomonas* increased slightly (Figure 7B). However, at 8 days after the second addition of *Pseudomonas*, the proportions of *Pseudomonas* and *Yersinia* increased sharply in the epidermal mucus flora. The proportions of *Yersinia* and *Pseudomonas* seemed to compete against each other (Figure 7B). At the end of the experiment, the proportion of these genera decreased, and the composition of the bacterial flora had changed compared to the initial phase (Figure 7C).

We next examined the antimicrobial bacteria on the fish epidermis and in the rearing water (Figure 7D); initially, the proportion of the antimicrobial bacteria was low in both (0.18% and 0.48%, respectively). However, after first administration of those bacteria, the proportion of the indigenous *P. parafulva* in the skin mucus increased. After the second addition of *Pseudomonas*, the proportions of *P. koreensis* and *Y. ruckeri* in the skin mucus began to increase, while the proportion of *P. parafulva* decreased. This suggests that the composition of the epidermal bacteria was disturbed by bacteria present in the water. When the occupation of *Pseudomonas* and *Yersinia* occurred, the proportions of *P. koreensis*, *Y. ruckeri*, *P. marginalis*, and *P. protegens* clearly increased. By the final day of the experiments, the proportions of *P. koreensis*, *P. marginalis*, and *Y. ruckeri* decreased, but the proportion of *P. protegens* continued to increase, and the proportion of the indigenous *P. parafulva* began to increase again (Figure 7D).

In the rearing water, the proportion of *P. protegens* clearly increased 3 days after the first dose, whereas an increase was not observed after the second dose. The proportions of the other *Pseudomonas* species and *Yersinia* in the water remained low in contrast to increases in the skin mucus (Figure 7D). These results demonstrated that antimicrobial-substance-producing bacteria in the water promote a transition of the fish epidermal mucus bacterial flora by migrating to the epidermal mucus and occupying it, at least temporarily.

## 4. Discussion

The epidermal mucus bacterial flora of fish is altered by environmental perturbations, such as changes in water quality [11,13,15,16,17,18]. However, it has been unclear whether the specific bacteria in the environment, other than specific pathogens infecting fish skin, affect the composition of the epidermal mucus bacterial flora. In this study, we found that bacteria producing antimicrobial substances and also present in the environment besides the bacteria pathogenic to fish also trigger changes of the epidermal mucus bacterial flora in a rainbow trout reared in flow-through aquaculture.

The monthly change in the composition of the epidermal bacterial flora in flow-through aquaculture would be caused partly by seasonal changes of the environmental conditions. Such a fluctuation seemed to be observed in Figure 3B as the spread of data in the “main group”. On the other hand, the large changes beyond the spread of the main group data were caused by the occupancy of the epidermal flora by certain bacteria (Figure 1 and Figure 3). The major bacterial species that occupied the epidermal mucus bacterial flora of rainbow trout on the fish farm belonged to the genera *Pseudomonas*, *Vibrio*, *Flavobacterium*, *Acinetobacter*, *Yersinia*, *Methylobacterium*, *Aeromonas*, and *Moraxella* (Figure 1). These genera, except for *Moraxella* and *Methylobacterium*, have been reported to include species pathogenic to fish [22,26,27,28,29,30]; however, the occupation of these genera on the epidermis is not always related to outbreaks of diseases, excluding the case of *Flavobacterium*. Thus, most of these skin-occupying bacteria are suggested to be nonpathogenic for rainbow trout.

The proportion of such mucus-occupied bacteria in river water or rearing water was independent of the proportion on the epidermis—and was often lower than on the epidermis (Figure 2; Appendix A), which indicates that their growth depends on the fish epidermal mucus to which they attach and grew well on.

Among the bacteria temporarily occupying the fish epidermal mucus, we concentrated on the antimicrobial bacteria, because treatment with antibiotics triggered the occupation of the water-borne pathogens on the fish epidermis [21]. We isolated *Pseudomonas* spp. and *Yersinia* spp. with antimicrobial-substance-producing capacity from fish epidermal mucus and showed that they exist both in the skin mucus and/or in the rearing water, but are nonetheless related to the occupation and disturbance of the mucus bacterial flora by specific bacteria (Table 1; Figure 5, Figure 6 and Figure 7). Our results suggest that the antimicrobial substances produced by them, whether the bacteria are present in the skin mucus or in rearing water, influence the composition of the fish epidermal mucus bacterial flora. By contrast, *Acinetobacter* spp., *Vibrio* spp., *Flavobacterium* spp., and *Moraxella* spp., which are also observed to occupy the epidermal mucus flora, were hardly as evident based on the screening (Table 1). The occupation of the flora with these bacterial species often occurred with or after the disturbance of the epidermal mucus flora by *Yersinia* or *Pseudomonas* (Figure 1), indicating that their occupation occurred in the transition phase of the epidermal mucus bacterial flora along with the disturbance by antimicrobial bacteria. However, since the type of culture medium used in this screening would affect the isolation of some bacteria, we should consider the possibility that other antimicrobial bacteria exist on the epidermis or in the rearing water. In fact, another study reported a case where *Flavobacterium* spp. and *Bosea* spp., which produce antimicrobial substances, were isolated from farmed rainbow trout epidermis [32].

In our experiments designed to test the artificial addition of antimicrobial bacteria *Pseudomonas* spp. and the fish pathogen *Y. ruckeri* to the rearing water, the epidermal mucus bacterial flora of rainbow trout was temporarily occupied/disturbed by them, and the composition of the epidermal flora changed from that initially present (Figure 7). In this occupation, disturbance, and transitional phase, antimicrobial bacteria seemed to be removed from the fish epidermis after their temporary occupation, and the proportion of other bacteria in the flora started to increase (Figure 7B). The latter bacteria also included antimicrobial bacteria, such as the indigenous *P. parafulva* in the skin mucus and *P. protegens* in the water (Figure 7D). This rapid change of the occupancy rate might have occurred because these bacteria are closely related to pathogenic fish bacteria and, therefore, might be easily attacked by the host’s immune system protecting the skin [1,3]. The immunoglobulin T secreted in the mucus (sIgT) is demonstrated to have both the functions of pathogen elimination and protection of beneficial skin microbes [33]. The change in the proportion of a particular bacterial species, as their occupancy rate decreased, might have occurred by translocation of bacteria in the water to the fish epidermis or because of the growth of the bacteria on the epidermis; therefore, the composition and abundances of the bacterial species in the flora would likewise rapidly change. On the fish farm, in March 2020, a temporary increase of the proportion of *Flavobacterium* coincided with an increase in the proportion of *Pseudomonas* in the epidermal mucus; an increase in the occupancy rate of *Vibrio* spp. at the beginning of 2020 followed the occupancy of *Yersinia* spp. in December 2019. These situations appear to be examples of the translocation of bacteria in the water (Figure 1), as *Vibrio* was first detected in a low proportion in river water in January, and *Flavobacterium* increased in the river water before the occupation was observed (Appendix A).

## 5. Conclusions

From the results of this study, we demonstrated one of the mechanisms causing fluctuations in the epidermal mucus microflora of rainbow trout in a flow-through aquaculture system as being associated with the bacteria in the environment. Especially, antimicrobial bacteria could cause the fluctuations of the proportions of bacterial genera in the fish epidermal mucus by disrupting that flora with antimicrobial substances and trigger the transition of the fish skin bacterial flora by promoting the translocation of bacteria in the water and the growth of the indigenous mucus bacteria (Figure 8).

Nonpathogenic bacteria that produce antimicrobial substances are often envisioned for application as biological control agents to prevent pathogenic infections in a variety of fields, with many studies conducted on screening such bacteria as part of disease-control methods. In the field of fisheries, many bacteria that produce antimicrobial substances have been obtained by screening, and their preventive effects regarding fish infections have been reported [34,35,36]. One mechanism of the effect relies on antagonizing the growth of the pathogenic bacteria by producing antimicrobial substances. Besides providing information on fish epidermal mucus bacterial flora, our findings provide a possible strategy for disease control using these antimicrobial bacteria, which appear to “remodel” the bacterial flora by disturbing them. Bacteria producing antimicrobial substances can also affect epidermal mucus bacteria aside from pathogenic bacteria; this triggers adhesion of the symbiotic bacteria in the water to the epidermis or the growth of epidermal mucus symbiotic bacteria in the bacterial flora. Alternatively, these disruptive effects could also lead to attachment by waterborne pathogens to the fish epidermis and to an increase in the number of opportunistic pathogens existing on the epidermis. Therefore, if such bacteria are to be used as biological control agents, it might be necessary to remove the pathogens with water and supplement the symbiotic bacteria at the same time to achieve higher efficacy.

## Figures and Tables

**Figure 1 biology-11-01249-f001:**
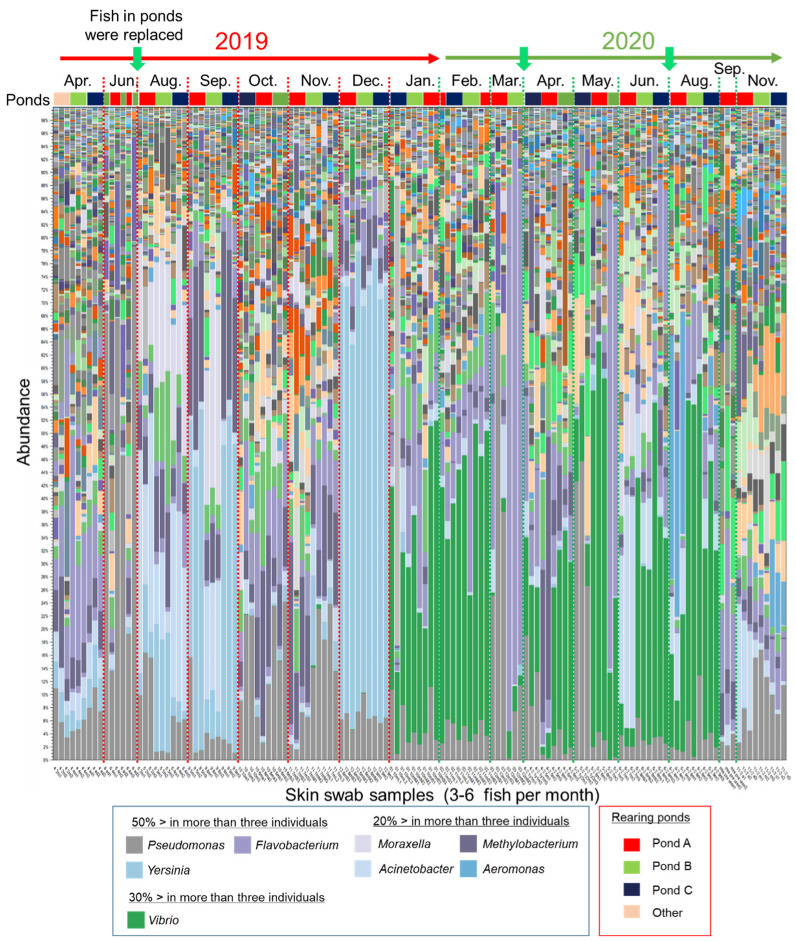
Bacterial composition in the epidermal mucus of rainbow trout reared in a flow-through aquaculture pond. The stacked bar chart shows the composition and abundance of operational taxonomic units (OTUs) (top-100, at the genus level) in epidermal mucus samples. The samples were collected every 1 to 2 months in 2019 and 2020. Green arrows indicate when all the fish in the ponds (Ponds A, B, and C) were replaced. The more abundant bacterial genera (e.g., >50% in more than three individual fish) are indicated in the figure legend.

**Figure 2 biology-11-01249-f002:**
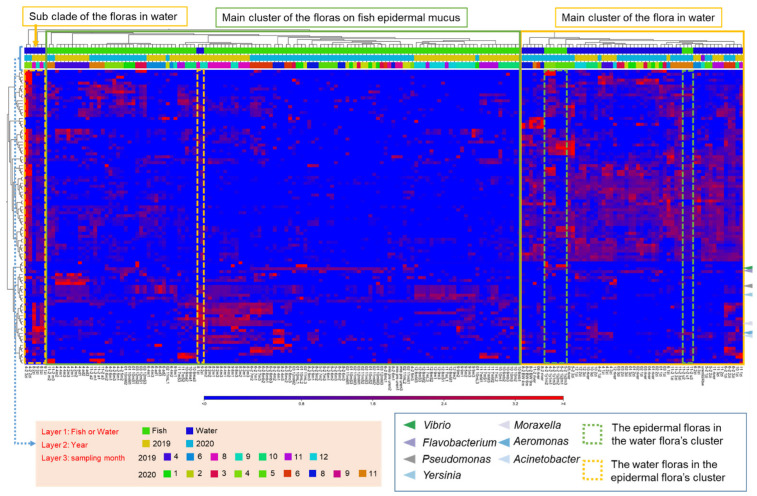
Specific abundance changes in OTUs obtained from fish epidermal mucus, independent of their abundance in the water or other sampling conditions. The composition of the bacterial flora from mucus samples and rearing water samples was classified based on the relative abundance of the top-100 OTUs shown in a clustered heatmap. The green frame represents the main clusters formed by the mucus samples, and the yellow frame represents the main clusters formed by the water samples. The dashed frames represent samples classified in another cluster than the main cluster. Arrowheads indicate the OTUs related to the skin-occupied bacteria. Sampling conditions (fish sample or water sample, sampling year, sampling months) are represented by bars with different colors, as defined in the legend.

**Figure 3 biology-11-01249-f003:**
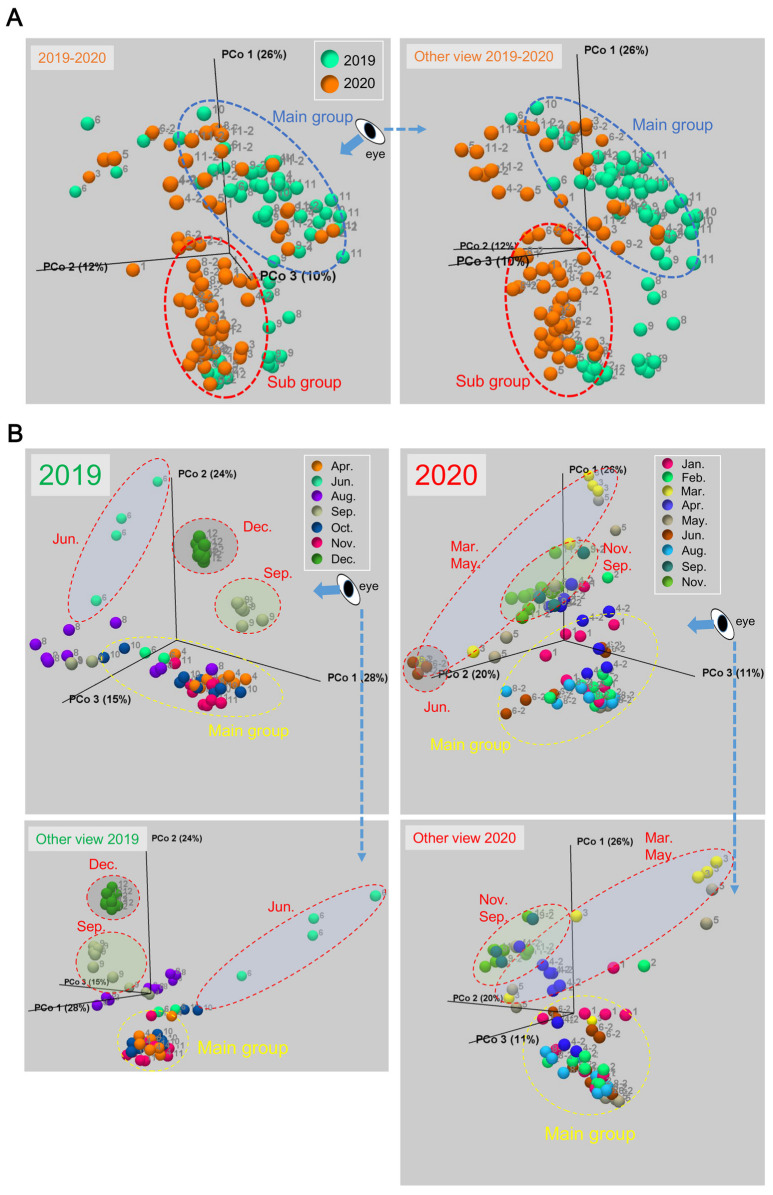
Beta-diversity of the skin mucus bacterial flora of rainbow trout in flow-through aquaculture. (**A**) For β-diversity, the weighted UniFrac distance was calculated using all data from both years (2019 and 2020), and the data are visualized in a 3D-PCoA plot. The blue dotted circles represent the main group of the skin mucus bacterial flora, and the red dotted circles represent the subgroup. The number next to each dot indicates the sampling month; the other view from the indicated direction marked “eye” and with arrows is also shown. (**B**) The β-diversity was calculated using the data for each year separately. The yellow dotted circles represent the main group of the flora, and the red dotted circles represent groups that clearly differed from the main group. The number next to each dots indicates the sampling month; the other side view from indicated direction is also shown.

**Figure 4 biology-11-01249-f004:**
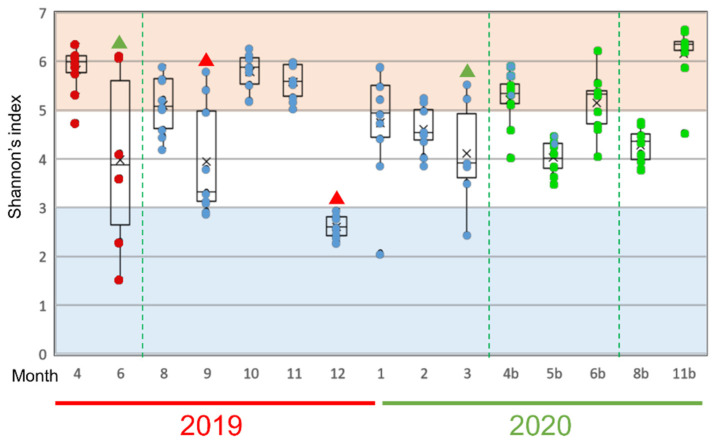
Alpha-diversity of epidermal mucus microflora of rainbow trout in a flow-through culture system. Shannon’s diversity index was calculated by rarefaction analysis, using 2.400 reads sampled randomly from each mucus sample. The results from 6–9 individual fish in each sampling month are represented by box plots; dots inside the boxes represent data from individuals, and the same color indicates fish reared from the eggs collected in the same fiscal year. Green triangles indicate months that the disturbance of the flora by specific bacteria was observed. Red triangles indicate the months that the occupation by specific bacteria was observed. The dotted green lines mark when fish were replaced on the farm.

**Figure 5 biology-11-01249-f005:**
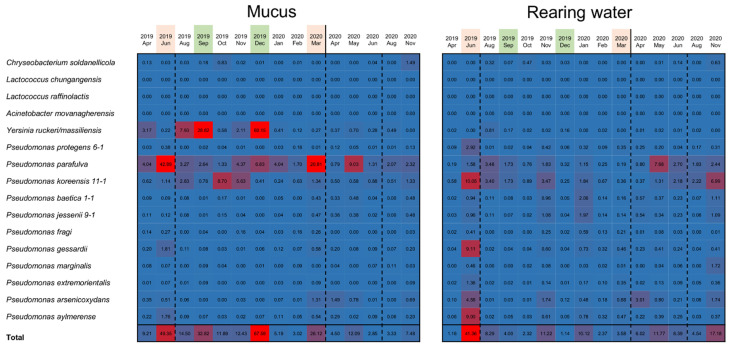
Confirmation of antimicrobial bacteria species in the epidermal mucus and rearing ponds of rainbow trout. Sequence reads randomly sampled from the data of each month (skin mucus samples and water samples) were mapped on the sequence list of the 16S rDNA V1–V2 region of antimicrobial bacteria isolated from the skin mucus samples. The number of reads mapped on the sequence for each antimicrobial bacteria was counted, and the proportion (%) of the mapped reads for each bacterium is presented as a heatmap (number of reads per total reads mapped on each bacterium). The months of observed disturbances of the flora are represented by pink, and occupation of the flora is represented by green.

**Figure 6 biology-11-01249-f006:**
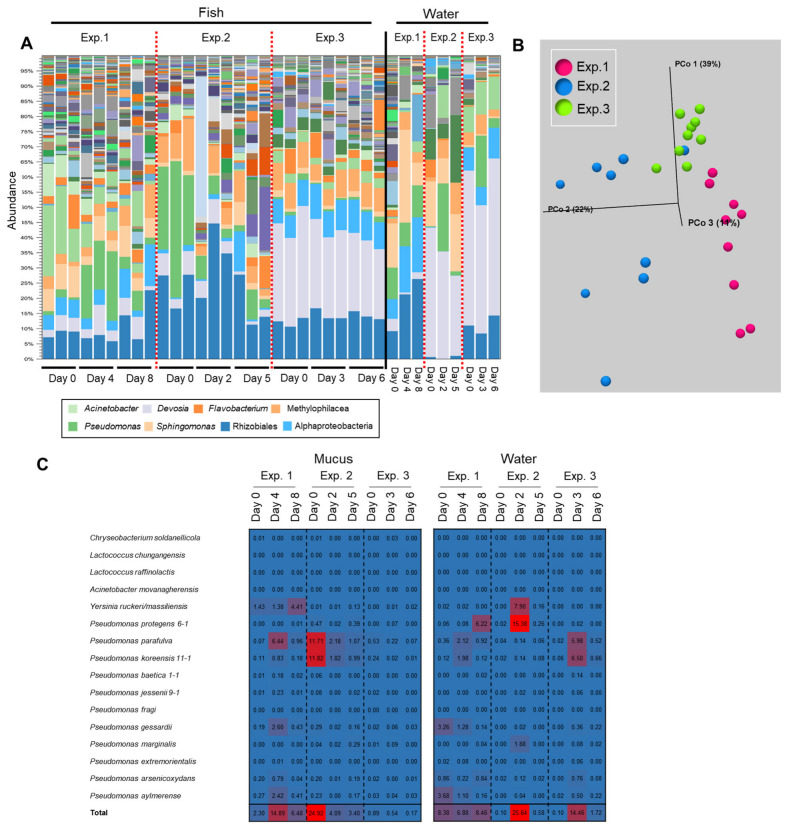
Results depicting a transition of the epidermal mucus bacterial flora of rainbow trout reared in a laboratory aquarium with a recirculating system. (**A**) The stacked bar chart shows the abundance of major bacterial flora in the epidermal mucus and rearing water, in an experiment performed three times (Exp. 1–3). (**B**) Beta-diversity measured by weighted UniFrac distance using fish skin mucus samples from Exp. 1, 2, and 3. (**C**) Reads mapped on the antimicrobial bacterial 16S rDNA sequences; the proportions of mapped reads from each day are depicted in a heatmap.

**Figure 7 biology-11-01249-f007:**
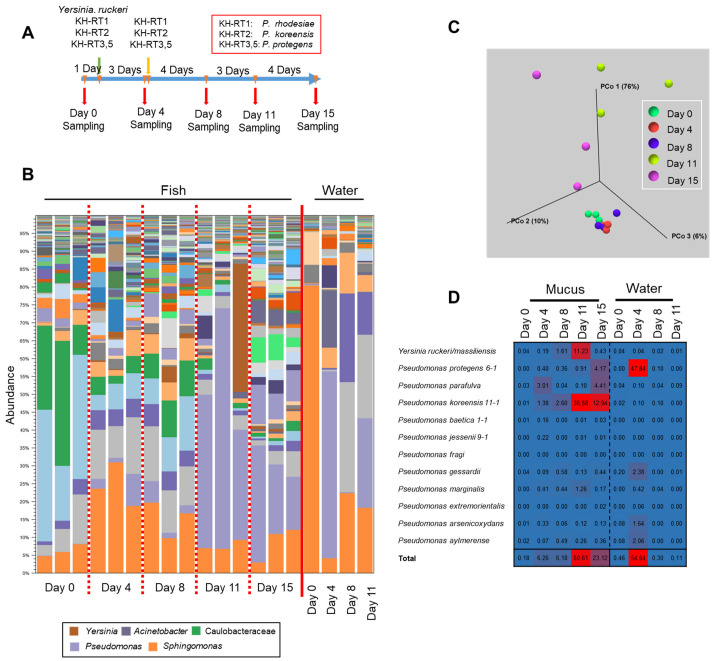
Transition of the epidermal mucus bacterial flora of rainbow trout by artificial administration of the antimicrobial *Pseudomonas* and *Yersinia* in laboratory aquaculture. (**A**) The schedule of the administration of the microbes and sampling of the fish and rearing water. (**B**) Stacked bar chart showing the abundance of major epidermal mucus bacterial flora of the rainbow trout before and after adding *Pseudomonas marginalis* KH-RT1, *P. koreensis* KH-RT2, *P. protegens* KH-RT3, and KH-RT5 and *Yersinia ruckeri* NCTC12266 to the rearing water. The *Pseudomonas* spp. were administrated twice during the experiment. (**C**) Beta-diversity, determined by the weighted UniFrac distance, calculated for the skin mucus samples obtained during the experiment. (**D**) Mapping reads of 16S rDNA sequences to the antimicrobial bacteria *Pseudomonas* and *Yersinia*; the proportion of mapped reads for each day are presented as a heatmap.

**Figure 8 biology-11-01249-f008:**
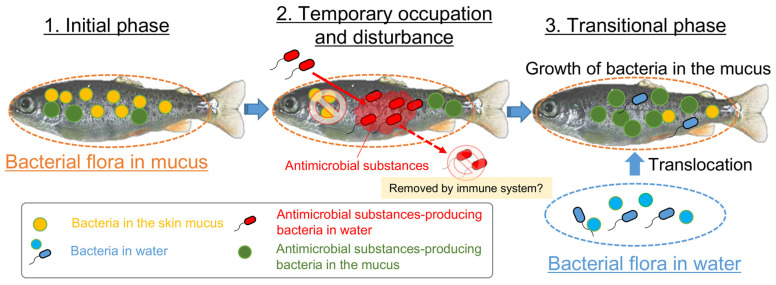
The proposed mechanism of the occupation, disturbance, and transition of the fish epidermal mucus bacterial flora by the bacteria with antimicrobial activity. The antimicrobial bacteria in the mucus and water contribute to the frequent change of the epidermal mucus microflora by occupying and disrupting the flora through the production of antimicrobial substances; this might cause a fluctuation in the proportions of epidermal mucus bacteria species and the transition of the flora by promoting the translocation of bacteria in the water and the growth of the indigenous mucus bacteria.

**Table 1 biology-11-01249-t001:** The number of isolated bacterial clones with antimicrobial activity identified in the skin mucus layer of rainbow trout in a flow-through aquaculture system.

Identified Bacteria (16s rDNA V1–V3)	Percent Identity against Nucleotide Database (%)		2019 (Month)							2020 (Month)			
	(16S_Ribosomal_ RNA)	(nt)	4	6	8	9	10	11	12	1	2	3	Total
*Acinetobacter* sp./*movanagherensis*	95.62	96.12 (*Acinetobacter* sp)						1					1
*Chryseobacterium soldanellicola*	97.51	97.67					1						1
*Lactococcus chungangensis*	99.74	99.74							1				1
*L. raffinolactis*	99.48	100.00				2							2
*Pseudomonas arsenicoxydans*	99.74	100.00										2	2
*P. aylmerense*/sp.	99.48	100.00 (*Pseudomonas.* sp.)										1	1
*P. baetica*	99.23	100.00			1					1		4	6
*P. extremorientalis*	99.74	100.00		3	1							1	5
*P. fragi*	99.74	100.00						1					1
*P. gessardii*	99.74	100.00		1									1
*P. jessenii*	99.74	100.00				1							1
*P. koreensis*	99.49 (*P. baetica*)	100.00		3	1	3		4				1	12
*P. marginalis/rhodesiae*	99.48	100.00	2										2
*P. parafulva*	99.48	99.74			5								5
*P. protegens*	99.74	100.00		1									1
*Yersinia ruckeri/massiliensis*	99.49	100.00						3	1				4
unidentified	-	-							2				2

Numbers within cells are the number of obtained clones. The green background denote *Pseudomonas* species. (): the most similar species in the database.

## Data Availability

The data presented in this study are openly available in the DDBJ Sequenced Read Archive (DRA) under the accession numbers DRX364235–DRX364335 and DRX363766–DRX363792.

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
