# Peer review of "Perturbation by Antimicrobial Bacteria of the Epidermal Bacterial Flora of Rainbow Trout in Flow-Through Aquaculture"

_biology, 2022, doi:10.3390/biology11081249_

Round 1

Reviewer 1 Report

The manuscript examined the epidermal mucus bacterial community of rainbow trout reared in flow-through aquaculture over 2 years, analyzed the effect of the abundance of bacteria in water on that of epidermal mucus bacteria, and explored the effect of the addition of antimicrobial bacteria to the rearing water on transition in the mucus bacteria community composition. Then, they provided a novel information on factors influencing the composition of fish skin bacterial flora, which is applicable for controlling fish disease by using beneficial bacteria in aquaculture. These results were encouraging. However, there are several points that need further explanation.

1. Please explain why do need a two-year examination of the epidermal mucus bacterial community of rainbow trout. As shown in Figure 1, the composition of microbial communities in fish epidermal mucus varied greatly at different periods of time. How do explain these changes?

2. Several fish rearing experiments were carried out, and it was necessary to supplement the fish culture facilities,water volume, size of fish, breeding density, mortality, etc. And the size of sampling fish collected was inconsistent across several trials, even within the same trials, the size of fish was inconsistent.

3. Figure 2:Too much information in Figure 2, too small to see clearly.

4. Line 337-344:Why do not use the same batch of fish? How do we distinguish them when sampling them? What is the effect of change of the test fish on the test results?

5. Line 358:Here were 48 strains, but there was 46 strains in the following data and in Table 1, and 50 strains in Table S3.

6. Line 408-421 and Figure 6:Explain the differences of fish epidermal mucus bacterial community between the three experiments, and what caused this difference?

7. Line 430:Figure S3 showed that pH value of experiment 2 was significantly low. In Experiment 2, the colony structure changed the most. Was it related to pH?

Reviewer 2 Report

This manuscript describes long-term change of the bacterial composition in skin mucus of rainbow trout reared in flow-through aquaculture as well as in those reared in a laboratory aquarium. Furthermore, they obtained antimicrobial bacteria from the skin mucus of rainbow trout, and utilized them as a potential trigger of compositional change in the epidermal mucus bacterial flora. Microflora in the fish epidermal mucus layer is expected to be utilized for disease control in aquaculture. However, there is little basic research to understand it yet. The authors collected a sufficient amount of basic data about microflora in fish skin. This manuscript will make an important contribution to the field of aquaculture microbiology. However, there are several points that might be taken into consideration. First, this manuscript contains four big experiments. The description is not well organized and it is hard to understand everything on first reading. For example, there is no description about the details of a flow-through system in the method section, so it is confusing. Concerning the conclusion and Figure 8, the main criticism is that they did not use bacteria without antimicrobial activity. The authors need to show that such a bacteria does not cause any disturbance when added similarly to thoroughly prove their hypothesis.

Minor comments are as follows.

Materials and Methods section: Please add description of the flow-through aquaculture system. 2.4. can be improved to clearly show that two sets of experiments were conducted.

L195: Please clarify that five strains were added at first, and only four were used for the second addition.

Fig. 1: Flow through system is very complicated. New fish were introduced into pondA, but not pondB and C. It is better to indicate the pond name in Fig. 1. Maybe it is shown at the bottom of the figure, but the resolution is not enough. How many days after the renewal of fish did you take samples? This point is important.

L243: “rearing water, river water” Does Fig. 2 contain data of both rearing water and river water?

Figure 5: It would be interesting to show the correlation diagram of the proportion of each strain between mucus and water.

Were Flavobacterium OTUs dominated in March and May 2020 close to F. psychrophilum? 

Line 117: 0.45 µm filer is not common for analysis of environmental bacteria, and needs to be explained.

L121: Briefly describe the name of the method of DNA extraction.

Fig6A: Why is data of water not shown here?

Fig. 7A: It is better to indicate the species name of KH-RT1, 2, 3.

Discussion: The meaning and definition of “disturbance” and “occupation” in this study is not clear, and needs more explanation.

L355: please indicate percent similarity between the isolates and the closest species.

L391: Again, the readers may want to see the correlation diagrams between the proportions in mucus and water.

L413: correct “ntimicrobial”

Conclusions: Bacteria that do not produce antimicrobial substances were not examined in this study. So the authors should be cautious and speculative conclusions should be avoided.
